# A Retrospective Observational Study Assessing the Clinical Outcomes of a Novel Implant System with Low-Speed Site Preparation Protocol and Tri-Oval Implant Geometry

**DOI:** 10.3390/jcm11164859

**Published:** 2022-08-18

**Authors:** Giacomo Fabbri, Tristan Staas, Istvan Urban

**Affiliations:** 1Studio Odontoiatrico Specialistico Ban Mancini Fabbri, Via del Porto 17, 47841 Cattolica, Italy; 2Staas & Bergmans, Schubertsingel 32, 5216 XA ‘s-Hertogenbosch, The Netherlands; 3Urban Dental Center Kft, Pitypang Street 7, 1025 Budapest, Hungary

**Keywords:** implant site preparation, tri-oval implant, implant survival, bone remodeling

## Abstract

A novel, biologically friendly implant concept system introduces low-speed (50 rpm) site preparation instruments used without irrigation and a tri-oval, tapered implant designed to reduce stress on cortical bone without sacrificing mechanical stability. This retrospective, observational, multicenter study (clinicaltrials.gov NCT04736771) collected data from consecutive patients treated with at least one novel concept system implant to evaluate clinical outcomes after 1 year in function. The primary endpoint was a marginal bone level change (MBLC) from loading to 1 year, and secondary endpoints included implant survival and clinician feedback. Ninety-five patients (54 women and 41 men, mean age: 58 ± 12 years) were treated with 165 implants. For 94.5% of implants, site preparation was performed in two steps. The mean follow-up from implant insertion was 1.8 ± 0.2 years. Mean MBLC from implant loading to 1-year follow-up was +0.15 ± 0.85 mm (*n* = 124 implants). At the last follow-up, the implant survival rate was 98.0%. Clinician satisfaction with the novel concept system was high. The novel concept system offers an easy-to-use implant placement protocol, with most implants placed using two steps. The minimal bone remodeling and high survival rate observed across a variety of indications and treatment protocols demonstrate broad versatility and confirm the clinical benefits of this biologically friendly innovation.

## 1. Introduction

The search for innovative materials and techniques able to improve treatment outcomes while simultaneously reducing morbidity and shortening the times required for surgical procedures and healing represent intense research topics in dentistry, with an increased focus on reducing extraction-related trauma and improving the implant site environment to promote implant integration [1]. Recent changes in the landscape of modern implant dentistry, as well as in patient expectations, have redefined the needs and requirements for implant-supported restorations. In the past, dental implants were placed by a few highly trained and experienced clinicians. However, more recent implant placements are increasingly being performed by general practitioners, with those placing 30–50 implants per year representing the fastest-growing segment. Patient expectations are also changing, with patients desiring shorter time to teeth, fewer visits, and implants that are both functional and provide natural-looking esthetics [2]. Furthermore, clinicians place implants in increasingly challenging conditions, such as sites with limited bone volume or fresh extraction sockets [3,4].

Successfully integrated dental implants form direct bone-to-implant contacts without intervening in non-osseous or connective tissues [5,6]. This process is influenced by both the implant characteristics, including geometry and surface features, and the preparation of the implant site [7,8]. A non-traumatic surgical preparation is essential for achieving the successful integration of dental implants, contributing to the long-term stability of these devices [1]. Conventional drill protocols used to prepare the implant site typically depend on the use of a series of high-speed drills (faster than 800 rpm) with increasing diameters that can introduce surgical trauma due to bone overheating, cortical compression, and damage to the trabecular micro-architecture [9,10,11,12]. During osteotomies, irrigation is typically used to both lubricate and cool the drill and the surrounding bone. However, an unwanted effect of irrigation is the removal of the resultant bone coagulum, a mixture of bone chips, connective tissue stroma, and blood, which has been found to have the osteogenic potential [10,13,14,15].

Alternative techniques have been developed to overcome the limitations associated with conventional drilling approaches used for implant site preparation, including osteotomes, Er:YAG lasers, osseodensification burs, and piezoelectric devices. However, a meta-analysis and systematic review found no significant improvements associated with the use of any of these methods compared with conventional drills for bone-to-implant contact or implant survival [16]. The newly developed N1 Concept System (Nobel Biocare AB, Göteborg, Sweden) implant site preparation protocol first utilizes a pilot drill to generate a pilot osteotomy (Figure 1), similar to conventional site preparation processes. The pilot osteotomy is then enlarged using novel osseoshaping instruments named OsseoShapers (Nobel Biocare AB, Göteborg, Sweden) that operate at the low speed of 50 rpm (Figure 1). These osseoshaping instruments are designed to create larger bone chips than conventional drills and rotate in the reverse direction when being removed from the osteotomy, which deposits the bone chips into the hole [10]. The low speed produces a temperature below that associated with osteocyte necrosis, requiring no irrigation [17,18,19], and the resultant bone coagulum contributes to bone formation, improving implant integration and stability [10].

In rat models comparing conventional site preparation using a high-speed drill and irrigation with site preparation using miniaturized osseoshaping instruments and no irrigation [10,20], the conventional drill-based protocol was found to create a smooth, glassy surface, and irrigation resulted in the removal of autologous bone chips. By contrast, the osseoshaping protocol resulted in a heteromorphic surface coupled with the retention of both collagen and bone chips, which was associated with reduced apoptosis and early bone regeneration, suggesting that the use of osseoshaping instruments contributes to the formation of a biologically friendly environment that supports osseointegration [10,20]. Another advantage of the low-speed osseoshaping tools is the ability to use these tools to evaluate the quality of the implant site based on the torque required for site preparation. A recent study [18] examining the use of the osseoshaping tools in bone surrogate materials, mini pigs, and human patients found strong, linear correlations between the torque required for site preparation using the osseoshaping instruments and both bone density and implantation torque, allowing clinicians to evaluate the implant site during site preparation, without additional assessments. A recently published clinical case series examining the use of osseoshaping instruments to place fifteen implants in seven patients found that, in most cases, the entire procedure could be completed in only three steps: pilot drilling, osseoshaping tool use, and implant insertion [20]. The reduced time, noise, and vibration associated with the use of osseoshaping tools also had positive impacts on patient comfort [20].

In addition to the characteristics of the implant site, the implant geometry and surface features, including composition, hydrophilicity, and texture, contribute to both long- and short-term implant success and can determine the rate and quality of osseointegration [8,21]. A tri-oval, tapered implant has been designed for use with the N1 Concept System. The tri-oval design provides areas of high strain at the maxima, providing initial mechanical stability, whereas the low strain regions at the minima are pro-osteogenic, resulting in faster bone growth that provides secondary stability [22]. The novel implant also features a gradually anodized surface, with increasing surface roughness from the collar to the apex to promote osseointegration [21]. The implant–abutment interface features a tri-oval conical connection that is designed to be self-centering, to lock the abutment in place, and provide a tight connection with the abutment, limiting the space available for bacterial growth.

These aspects of the novel concept system combine to produce a biologically friendly implant system that minimizes tissue damage during the implant procedure and maximizes osseointegration to promote long-term implant success. The objective of this retrospective, multicenter study was to evaluate the clinical outcomes of this system after up to 1 year in function. The primary endpoint of the study was to test the hypothesis that the marginal bone level change (MBLC) from implant loading to 1-year follow-up using novel concept system implants is non-inferior to the MBLC observed using a variable-thread tapered implant system, based on historical data. Secondary endpoints included implant survival and success rates, safety evaluations, and clinician feedback on esthetic and functional outcomes.

## 2. Materials and Methods

### 2.1. Study Design

A retrospective, observational, multicenter study was conducted on consecutive subjects treated with the N1 Concept System (Nobel Biocare AB), who were followed for up to 1 year after loading. The objective of the study was to evaluate the osseointegration of the N1 Concept System implants and test the study hypothesis that the MBLC from implant loading to the 1-year follow-up associated with the use of the novel concept system implants is non-inferior to historical MBLC data for the variable-thread, tapered implant system (NobelActive, Nobel Biocare AB).

A retrospective chart review was conducted to identify all implant patients who were planning to receive the novel tri-oval implant (Nobel Biocare N1, Nobel Biocare AB) at one of three private dental clinics located in Hungary, Italy, and the Netherlands. All patients who met the inclusion criteria at the study sites were enrolled in the study. Inclusion criteria included age ≥18 years; signed informed consent and consent to data processing, according to local regulations; and received at least one N1 implant. No exclusion criteria were applied to ensure the assessment of the novel concept system under typical circumstances encountered in daily clinical practice.

All data collection procedures were conducted in accordance with the 2013 amendment of the Helsinki Declaration of 1964 for biomedical research involving human subjects. Each of the participating study centers obtained an ethics vote or a waiver by an independent ethics committee before commencing any study activities (Italy: Prot.2312/2021, I.5/46, the Netherlands: NL76793.041.21, Hungary: PhR12716/2021). The study was registered at clinicaltrials.gov (accessed on 1 July 2022) (NCT04736771) prior to patient enrollment. The participating clinics applied standard inclusion and exclusion criteria for treatment with dental implants [23]. The surgeries took place from 6 December 2018 to 9 April 2020.

Data extraction included demographic data, medical conditions, previous and concomitant medications, implant size, implant site, and surgical characteristics. Radiographs of the implant region at implant insertion, implant loading, and up to 1 year after loading (±3 months) were obtained. Bone quality and quantity were classified according to Lekholm and Zarb [23].

Each study subject was pseudonymized and assigned a study ID, and clinical and radiographic data were retrospectively collected via chart review, de-identified, and entered into an electronic data capture system. No personal/confidential information was recorded in the study database, and all radiographs were de-identified before analysis. All subjects included in the study signed the required consent forms, as per respective national regulations, before any data were collected.

### 2.2. Surgical Procedures

Patients were treated according to the standard dental care procedures at the participating clinics. Implants were placed into fresh extraction sockets, healing sockets (25 h to 12 weeks post-extraction), or healed sites in a single tooth, partial bridge, or full-arch bridge indications. Access to the surgical site was flapless, with a minimally invasive or full flap. The osteotomy formation protocol followed the manufacturer’s recommendations: after the application of the pilot drill (OsseoDirector, Nobel Biocare AB), the osteotomy was enlarged using the osseoshaping instrument (OsseoShaper 1, Nobel Biocare AB), operated at the low speed of 50 rpm and without irrigation. If the instrument was fully seated (i.e., reached the intended depth) without exceeding the maximum torque of 40 Ncm, implant placement followed. In cases in which the instrument could not be seated fully, the osteotomy was enlarged with the dense bone–optimized osseoshaping instrument (OsseoShaper 2, Nobel Biocare AB) operated at 50 rpm and without irrigation. If the instrument was fully seated without exceeding the maximum torque of 40 Ncm, implant placement followed. In cases in which the instrument could not be seated fully, the osteotomy was further enlarged with a dense bone drill operated at high speed and with irrigation. Implants had a regular platform, and their length was 9, 11, or 13 mm. Implants were placed in a 1-stage or 2-stage surgery, and loading occurred immediately (within 48 h after implant insertion), early (between 48 h and 3 months after implant insertion), conventionally (from 3 to 6 months after implant insertion), or delayed (more than 6 months after implant insertion). Restorations only utilized Multi-unit and On1 two-piece abutments (both Nobel Biocare AB) due to the limited prosthetic portfolio available at the time of patient treatment. After implant placement, all patients received recommendations regarding medication, oral hygiene maintenance, and diet according to the standard postoperative care procedures at each treating clinic.

### 2.3. Outcome Measures

Marginal bone levels (MBL) were assessed using all available radiographic examinations (intraoral periapical radiographs, orthopantomogram (OPG), and periapical sections from cone-beam computed tomography (CBCT)) collected at the time of surgery, at implant loading, and at 12 months after loading. Loading was defined as provisionalization or final prosthesis delivery, whichever occurred first. For accuracy reasons, periapical radiographs or periapical sections from CBCTs were preferred, while OPGs were used if other radiographs were unavailable.

Bone-height measurements were made by an independent radiologist (Dr. Agneta Lith, University of Gothenburg, Sweden). All radiographic images were assessed twice, with a 2-week interval between the first and second assessments. The intra-rater reliability score (within 0.5 mm) was 77% for all radiographs (*n* = 751), 74% for OPGs (*n* = 266), and 79% for periapical radiographs and periapical sections of CBCTs (*n* = 485). Bone levels coronal to the top of the implant were expressed as positive values, whereas those apical to the top of the implant were expressed as negative values. The MBLC was calculated from paired radiographs, with positive values representing marginal bone gain and negative values representing marginal bone loss. The radiologist also assessed the radiographs for “absence of radiolucencies,” a criterion used for implant success, with a yes/no decision [24]. Missing data were not imputed and not included in the evaluation. All loaded implants were included in the radiographic analysis, irrespective of the type of loading protocol.

The primary endpoint of the study was designed to test the hypothesis that the MBLC observed using the novel concept system implant is non-inferior to the MLBC observed using an implant system placed following a conventional drilling protocol. Specifically, the MBLC measured for N1 Concept System implants was tested for statistical non-inferiority using a one-sample non-inferiority test and a margin of 0.5 mm against the weighted mean of −0.48 mm calculated for 151 implants described in two published references [25,26].

Secondary endpoints included implant survival; implant success according to the van Steenberghe criteria [24]; the clinician’s assessment of the self-centering effect (at the placement of the abutment or the cover screw); and the clinician’s satisfaction with the esthetic and functional outcomes (on a scale of 0–10, with 0 representing unsatisfied and 10 representing completely satisfied). No overall evaluation of the soft tissue response was possible because the study centers documented soft tissue health by either using non-standardized, clinic-specific parameters or recording only those instances in which the peri-implant soft tissue had an unhealthy appearance. However, a statement on soft tissue health was obtained for all surviving implants. Finally, safety parameters, including all adverse events that occurred from implant placement up to 1 year after loading, were also recorded in the study database.

### 2.4. Statistical Analysis

Sample size calculation was based on the assumption that MBLCs (calculated as the change in MBL measurements between implant loading and 1-year follow-up) would follow a normal distribution, with a power of 0.9 and a confidence level (alpha) of 0.05. With a conservative approach to ensure sufficient power for the primary endpoint and the assumed standard deviation derived from a previously published study [27], the required sample size for this study was determined to be *n* = 65 subjects with readable X-ray pairs. The correlation between the torque of the osseoshaping tool and the final implant insertion torque was assessed by Pearson’s correlation coefficient. The impact of bone quality on the correlation was evaluated using an independence test [28]. Subgroup analysis to compare final implant insertion torque and marginal bone remodeling from loading to the 12-month follow-up by indication, site type, and loading protocol was performed using ANOVA with post hoc Tukey test for between-group comparisons. Statistical analyses were performed by a biostatistician in SPSS version 24.0 (IBM, Armonk, NY, USA) and R version 3.6.0 (R Foundation for Statistical Computing, https://www.R-project.org).

This manuscript follows the STROBE (STrengthening the Reporting of OBservational studies in Epidemiology) guidelines [29].

## 3. Results

This retrospective study evaluated the outcomes of the novel concept system in patients treated at three private practice dental offices. Of the 106 eligible patients, 95 patients who received 165 implants were included in the final analysis, as 11 patients either declined to participate or were unreachable. For 14 patients, no 1-year follow-up data were available; however, the data for these patients are included in the analysis for parameters assessed prior to the final follow-up. Figure 2 shows the flow of data collection and analysis.

At the time of surgery, the mean patient age was 58 ± 12 years (range, 23–86 years). Of the 95 patients, 43.2% were men, and 56.8% were women. Relevant medical history was reported for several patients and included nicotine use (20%), diabetes mellitus (5.3%), osteoporosis (3.2%), or a history of periodontitis (8.4%) or peri-implantitis/severe mucositis (5.3%). As detailed in Table 1, implant site and device characteristics comprised all indications; all locations in the jaw; healed, healing, and extraction sites; and a wide variety of loading protocols. The implant sites were prepared to utilize both flap and flapless procedures, and all types of bone quality and nearly all types of bone quantity were represented. Most of the implants (72.1%) were restored following the “one abutment, one time” concept, with the On1 base being placed at 46.7% of the implants and the Multi-unit Abutment being placed at 25.4% of the implants on the day of surgery. Two sample clinical cases from the study are shown in Figure 3.

### 3.1. Implant Placement Using the Novel Site Preparation Protocol

In 156 cases (94.5%), the implant was placed after using the novel osseoshaping instrument, whereas five implants (3.0%) were placed after using the dense bone-optimized osseoshaping instrument, of which two (1.8%) were in the maxilla, and three (5.5%) were in the mandible. The remaining four cases (2.4%) that required the use of the dense bone drill were all located in the mandible, at healed sites with advanced bone resorption. The final torque of the osseoshaping instrument was 22 ± 12 Ncm (range, 4–50 Ncm; *n* = 73), and the mean final implant insertion torque was 43.2 ± 16.5 Ncm (range, 9–75 Ncm; *n* = 165). Most implants (70.3%; *n* = 116) achieved an insertion torque above 35 Ncm. Table 2 lists the site preparation protocol, the final torque of the osseoshaping instrument, and the final implant insertion torque according to bone quality. Two implants (1.2%) failed to reach sufficient primary stability and were immediately replaced during the same surgery. In one case, a longer implant was used, and in the other case, another implant type with a wider diameter was used. The investigators were also asked whether the achieved primary implant stability was sufficient for immediate loading. Among the 97 implants for which a response was received, 89 implants were deemed to have sufficient primary implant stability to support immediate loading. For the 50 implants that were loaded immediately, the mean final torque of the osseoshaping instrument was 26 ± 6 Ncm (range, 14–36 Ncm; *n* = 17), whereas the mean final implant insertion torque was 51.3 ± 13.5 Ncm (range, 21–75 Ncm; *n* = 50).

In six cases, the clinician opted to apply a torque slightly higher than recommended. In four cases, the final torque of the osseoshaping instrument was <10 Ncm above 40 Ncm, and in two cases, the final implant insertion torque was <5 Ncm above 70 Ncm. The application of the osseoshaping instrument was the last step in osteotomy formation for all of these cases, and none of them required the use of either the dense bone-optimized osseoshaping instrument or the dense bone drill. None of these six cases were involved in implant survival failures or buccal plate fractures.

In a statistical analysis, the final torque of the osseoshaping instrument showed a moderate (R^2^ = 0.540) correlation with the final implant insertion torque (Figure 4). The correlation did not depend on the bone quality (*p* = 0.6906). With regard to indication, site type (healed vs. healing vs. extraction), or loading time, only loading time was associated with final implant insertion torque, where the torque of immediately loaded implants was higher than that of early loaded implants (*p* = 0.0334).

### 3.2. Outcome Measures

A total of 76 patients receiving 124 implants had readable paired radiographs for both implant loading and 1-year follow-up, which met the minimum sample size required according to the power analysis. The MBLC from the time of implant loading (1.45 ± 4.3 months after insertion, on average) to 1-year follow-up revealed a mean bone gain of 0.15 ± 0.85 mm (*n* = 124). This value was found to be non-inferior (*p* < 0.001) to the results reported for the historic implant group, in which the weighted mean MBLC at 1-year post-loading was −0.48 mm (*n* = 151). Indication, site type, and loading time had no statistically significant impact on MBLC at 1-year post-loading (all *p* > 0.05). The mean MBL was −0.79 ± 1.99 mm (*n* = 117) at implant insertion, −1.45 ± 1.43 mm (*n* = 129) at implant loading (radiographic baseline), and −1.24 ± 1.35 mm (*n* = 126) at 1-year follow-up. The mean marginal bone remodeling from implant insertion to 1-year follow-up (1.8 ± 0.2 years after implant insertion) was −0.53 ± 1.83 mm (*n* = 114).

At the 1-year follow-up visit, which occurred 1.8 ± 0.2 years after implant insertion, on average, implant survival was 98.0% (*n* = 141/144). Of the three failures, one implant had to be removed when attempting to disconnect the healing cap from the abutment base. The remaining two failures were recorded in a single patient, who was a heavy bruxer and refused to wear a recommended nightguard. The latter two failures occurred after final prosthesis delivery, 13 and 15 months post-implant placement.

In addition to the three non-surviving implants, three additional implants were categorized as non-successful, resulting in a 1-year success rate of 95.8%. Of these three additional cases, one was due to signs of implant mobility. The prosthesis was disconnected from the mobile implant, and the implant was judged as recovering by the treating clinician in a subsequent follow-up visit. In the second case, the clinical screw broke during placement of the abutment base, and the broken screw fragment was retrieved in a prolonged procedure. The MBL around this implant deteriorated after the retrieval procedure, and the implant remained unloaded to promote bone level recovery. In the third case, the implant was not connected to a functional prosthesis because it was part of a multi-unit restoration, and the other implant supporting this restoration (which was not a study implant) was showing complications and had to remain unloaded.

The initial clinical investigation protocol intended to collect soft tissue health parameters; however, due to the retrospective nature of the study, these data were not recorded in patient charts and, therefore, were unavailable for analysis. Nevertheless, according to the participating clinicians, all sites at surviving implants were judged as being healthy.

Overall, the evaluation of the clinicians’ assessments of the self-centering effect and of their satisfaction with the esthetic and functional outcomes showed positive results. The tri-oval conical connection was judged as self-centering in all cases (*n* = 165). Of the 140 assessed implants, clinicians rated their esthetic and functional satisfaction as very high, with respective mean scores of 9.1 and 9.4 on a scale of 0–10.

In total, seven adverse events were reported by the investigators during the time period from implant insertion up to 1-year post-implant loading. These events included three implant failures, two failures to reach sufficient primary stability at placement, one buccal plate fracture, and one case of implant mobility. No device deficiencies, serious adverse events, serious adverse device effects, or unexpected serious adverse device effects were reported.

## 4. Discussion

This retrospective clinical investigation was performed to evaluate the first clinical data on the osseointegration of the novel biologically friendly concept system and to test the hypothesis that MBLCs associated with the novel implants are non-inferior to those reported by previously published retrospective studies on the variable-thread, tapered implant system [25,26]. According to the primary endpoint analysis, the mean MBLC of +0.15 ± 0.84 mm (*n* = 124) from implant loading to 1-year follow-up was statistically non-inferior (*p* < 0.001) to the weighted mean MBLC of −0.48 mm (*n* = 151) from the two reference studies, allowing for a 0.5 mm margin. The standard deviation of 0.85 mm observed in this study was comparable to those documented by the two reference studies of 0.9 mm [26] and 1.3 mm [25]. The low MBL from implant insertion to 1-year follow-up and the bone gain observed from implant loading to 1-year follow-up meet the criteria for implant therapy success, which defines a successful implant (in terms of bone remodeling) as one with not more than 2 mm of bone loss from insertion to 1-year follow-up [30] and not more than 1 mm of bone loss from loading to 1-year follow-up [31]. Although in a statistical analysis of variables such as indication, site type, or loading time had no statistically significant impact on MBLC from loading to 1-year follow-up, these results should be interpreted with caution given the high heterogeneity of the sample in the study.

Secondary endpoints assessed in this study included implant survival, implant success, soft tissue healing parameters, and adverse events. The survival and success analyses were calculated using a conservative approach, considering only those subjects who completed the 1-year follow-up visit and all failures reported as adverse events. At the 1-year follow-up visit, implant survival and success rates were 98.0% and 95.8%, respectively. These findings are comparable to the survival rates reported for the two reference studies: 96.4%, with a mean follow-up of 1.3 years [26]; and 94.6%, with a mean follow-up of 1.1 years [25]. Of the three implant failures reported in the current study, two occurred after final prosthesis delivery in a patient with bruxism who refused to use a nightguard and likely resulted from overload. The third failure was due to a handling error, in which the healing cap on the abutment base could not be removed, requiring the implant to be replaced; thus, this failure does not represent a failure to osseointegrate or the loss of osseointegration.

Several factors are likely to account for the excellent marginal response and the high implant survival rate observed in this study. The clinical teams involved in the treatment of the study patients combine years of experience with dental implant therapy. The impacts of the individual surgeon and the prosthetic design on clinical success have previously been demonstrated, particularly for early failures [32,33]. Similarly, prosthetic design has been shown to affect early implant survival [32]. In addition, the novel biologically friendly system used in the study has been shown to promote enhanced osseointegration in pre-clinical evaluations by preserving bone viability in the osteotomy site, reducing heat-induced trauma, and retaining the osseous coagulum within osteotomies [10,13]. Furthermore, the tri-oval macroshape of the implant has also been demonstrated to have a positive impact on the bone response by providing relief to bone strain at its minima [22]. Finally, a large majority (72.1%) of the implants were restored following the “one abutment, one time” concept, which has previously been shown to benefit peri-implant tissue outcomes [34,35,36,37] and likely contributed to the excellent bone response observed in the current study.

A full evaluation of relevant parameters associated with the soft tissue response was not possible due to the retrospective nature of the study. The study centers documented soft tissue health by either scoring non-standardized, clinic-specific parameters or only recording those instances in which the peri-implant soft tissue had an unhealthy appearance. However, according to the treating clinicians, all sites at surviving and successful implants were judged as healthy, and none of the implant sites demonstrated evidence of substantial bone loss, which is a common consequence when the soft tissue surrounding the restoration suffers from poor health. In addition, the investigators rated the esthetics at the 1-year visit as highly satisfactory. Soft tissue conditions are typically key contributors to esthetic appearance; therefore, this level of high satisfaction with the appearance of the implants is indicative of good soft tissue health. Additional studies with standardized evaluation approaches are necessary to assess soft tissue outcomes with this novel implant system.

One buccal plate fracture was recorded in the current study. Such fractures were previously documented in a study of the predicate device, the variable-thread, tapered implant, in which four buccal plate fractures were recorded [38].

The broad range of patient and implant site characteristics, indications, and loading protocols included in this study provide a broad representation of the typical patient population encountered in daily clinical practice who receive treatment with dental implants. The study included both men and women located in three different countries and covered a broad age range. Both smokers and non-smokers were included, and other relevant medical pre-conditions, such as diabetes mellitus, osteoporosis, or history of periodontitis/peri-implant pathologies, were reported for several of these patients. The implant sites included all indications, all locations in the jaw, and both healed and extraction sites. The implant sites were prepared utilizing flap and flapless procedures, all types of bone quality were represented, and nearly all types of bone quantity were included. Only extreme resorption (class E) was not reported in this study; at the time of implant placement, short implants were not yet part of the available portfolio for the novel implant system, which may have contributed to the omission of class E patients.

The novel implant site preparation protocol used in the current study does not require prior assessment of bone quality classes to determine an optimized drill protocol. Instead, the decision to place the implant is solely based on whether the osseoshaping instrument can reach the full implant depth without exceeding the maximum torque of 40 Ncm. This feature removes the need to evaluate bone quality prior to implant placement and allows the clinicians to move through the subsequent steps of osteotomy formation until they achieve successful implant insertion. In the current study, the vast majority of implants (*n* = 165; 94.5%) were placed after a two-step implant site preparation, immediately following the use of the osseoshaping instrument. On average, the implants achieved high final insertion torques, with a mean insertion torque that would allow more than two-thirds of implants to be immediately loaded if desired by the treating clinician. In addition, the final torque recorded for the osseoshaping instrument correlated with the final implant insertion torque, independent of bone quality. These findings confirm that the novel concept system can result in stable implant placement without prior bone quality assessments, facilitating implant placement in soft or medium bone. This novel design is highly appreciated by patients because it reduces procedural time and is a gentle approach with less noise and fewer vibrations [20].

In six cases, the clinicians deviated from the Instructions for Use by applying torques slightly exceeding those recommended by the manufacturer. Torques up to 10 Ncm greater than recommended for the osseoshaping instrument are not expected to compromise the safety and effectiveness of the system, and none of the four cases associated with protocol deviation involved survival or success failures. Among the reported implant insertion torques, two implants exceeded the recommended maximum of 70 Ncm by up to 5 Ncm. Both osseointegrated successfully, and neither was involved in a survival or success failure.

The assessment of implant systems as they are used in daily clinical practice is important for clinical decision making, as clinicians must be able to provide all patients with satisfactory options. Patients who are selected for trials may be subjected to specific criteria that can bias outcomes or mask potential issues associated with excluded parameters. In addition, as observed in this study, some clinicians will opt to exceed the recommended system limits. Studies that evaluate implant systems as they are used in daily clinical practice allow all elements of the system to be evaluated under all possible circumstances, providing important information regarding the broad applicability of implant systems for clinicians to consider when treating patients.

The overall findings of this study indicate that the use of the novel concept system to prepare implant sites offers both patients and clinicians favorable functional and esthetic outcomes by creating a biologically friendly environment that supports the successful osteointegration of the novel system implant. The novel concept system offers a gentler site preparation process, which is appreciated by patients. In addition, the novel concept system removes the need to perform additional bone quality assessments, and it allows clinicians to predict implant stability based on the torque used during the site preparation protocol. The self-centering feature of the novel concept system implant was effective for all implants, further simplifying the implant loading process. The novel concept system was found to be non-inferior to a predicate device and was successful across a range of indications and patients, suggesting that it can be used in daily clinical practice.

The main limitations of this study arise from its retrospective design, the short follow-up, the highly heterogeneous nature of the study sample, and the relatively limited implant and prosthetic portfolio available at implant placement and prosthetic delivery. The retrospective study design limited the availability of information regarding soft tissue outcomes and the availability of periapical radiographs or periapical CBCT sections for assessment, with OPGs contributing to 35.4% of radiographic analysis. Future prospective studies remain necessary to assess the effects of the novel concept system on soft tissue healing. Although a 1-year follow-up post-loading provides insights into early failures and early bone responses, longer follow-up studies are needed to evaluate the long-term performance of this novel system. Finally, the lack of longer or shorter implants (the implants available in this study were 9, 11, and 13 mm long) and the limited prosthetic portfolio, which only included the two-piece abutments and multi-unit abutments, likely resulted in the exclusion of patients who would have been eligible for implant therapy.

## 5. Conclusions

This multicenter retrospective analysis reports the first data on the use of the novel N1 Concept System in daily clinical practice and demonstrates minimal bone remodeling from implant loading to 1-year follow-up, with a high implant survival rate, indicating successful osseointegration. The novel implant site preparation protocol allows implant placement after two steps at most implant sites, improving ease of use and limiting the number of surgical entries into osteotomies. The successful application of this concept system across a variety of indications and treatment protocols demonstrates the wide versatility of the novel concept system and confirms the clinical benefits of this biologically friendly innovation.

## Figures and Tables

**Figure 1 jcm-11-04859-f001:**
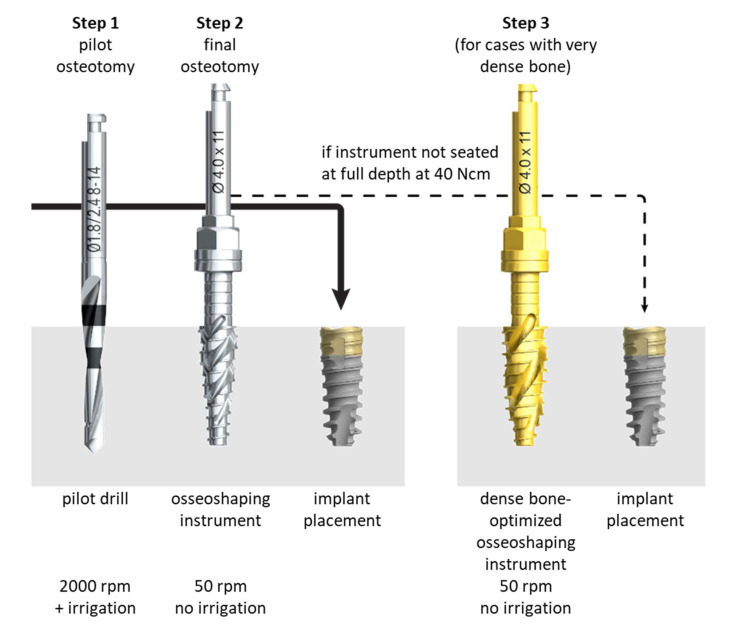
Osteotomy formation protocol. Implant placement follows a two-step site preparation in cases where the osseoshaping instrument can be fully seated without exceeding the torque of 40 Ncm (**left panel**). If the instrument cannot be seated, the osteotomy is further enlarged using the dense bone–optimized osseoshaping instrument using a torque of up to 40 Ncm (**right panel**). If needed, the dense bone drill is used (not shown).

**Figure 2 jcm-11-04859-f002:**
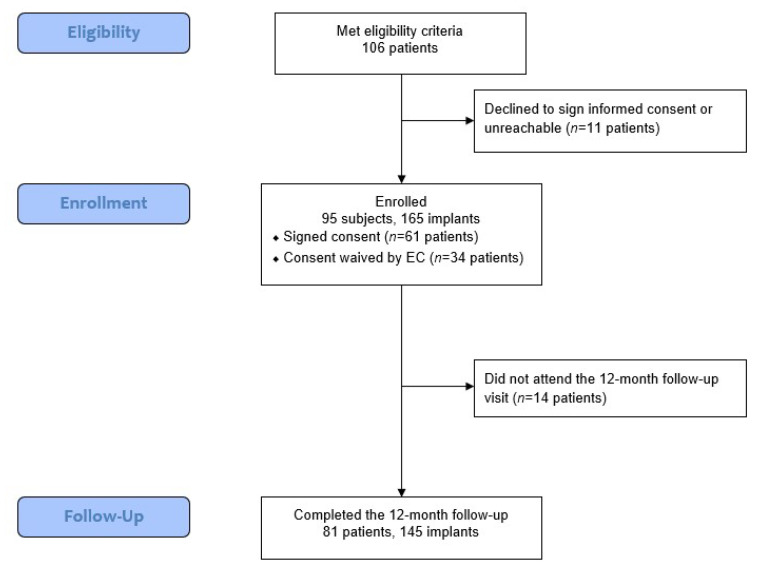
Data collection flow. EC, ethical committee.

**Figure 3 jcm-11-04859-f003:**
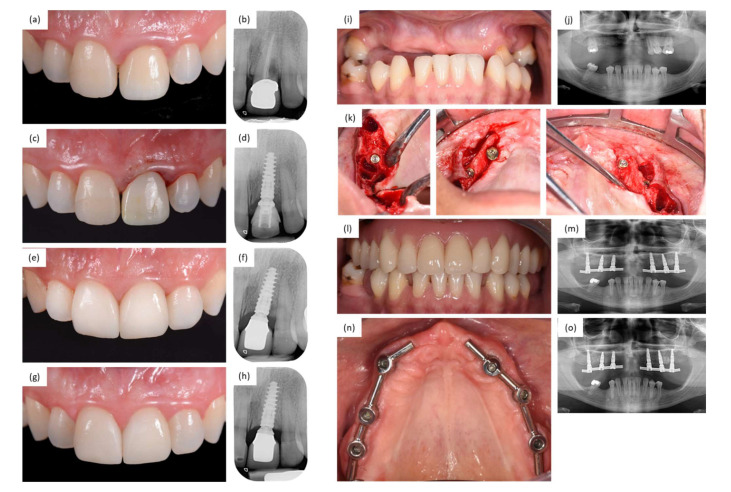
Sample clinical cases from the study. (**a**–**h**) A 44-year-old female patient was treated with a single implant to replace a failing maxillary incisor at FDI (Federation Dentaire Internationale) position 21 (**a**,**b**). The 13 mm long implant was inserted with the final insertion torque of 57 Ncm with a flapless approach, connected to an On1 base (0.3 mm NP (narrow platform)), and loaded immediately (**c**,**d**). The final prosthesis was delivered 2 months later (**e**,**f**). Note the stabilization of the marginal bone levels and the excellent soft tissue response at the 1-year follow-up (**g**,**h**). (**i**–**o**) A 31-year-old female patient with missing and failing dentition in the maxilla (**i**,**j**) was treated with 6 study implants to support a full-arch prosthesis (**k**–**m**). Implants were inserted at FDI positions 13 (healed site), 15 (healed site), 17 (extraction site), 23 (extraction site), 25 (healed site), and 27 (extraction site). Implant length and final insertion torque were 11 mm and 50 Ncm, 11 mm and 50 Ncm, 13 mm and 30 Ncm, 11 mm and 40 Ncm, 11 mm and 50 Ncm, and 13 mm and 40 Ncm. The implants were connected to multi-unit abutments and loaded immediately. The final prosthesis was delivered 5 months later. Note the excellent soft tissue recovery visible at 2 months post-implant insertion (**n**) and the stable marginal bone levels at the 1-year follow-up (**o**).

**Figure 4 jcm-11-04859-f004:**
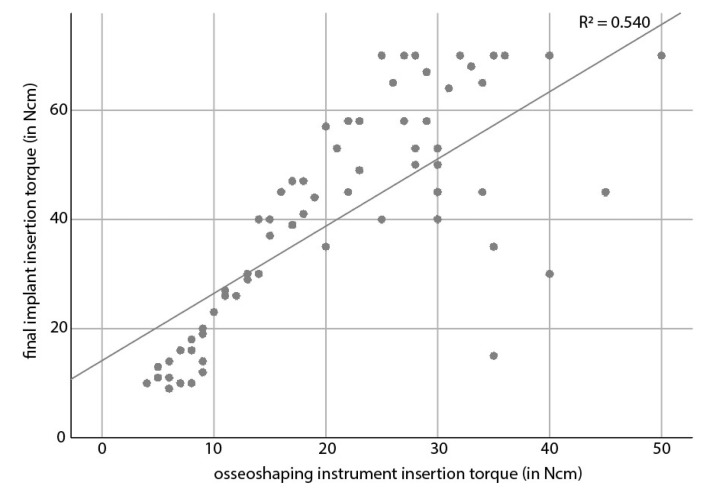
Correlation between the torque of the osseoshaping instrument and the final implant insertion torque.

**Table 1 jcm-11-04859-t001:** Baseline characteristics at surgery.

Implants		*n* (%)
		165 (100)
Implant position	Maxilla	110 (66.7)
	Incisors	25 (15.2)
	Canines	7 (4.2)
	Premolars	55 (33.3)
	Molars	23 (13.9)
	Mandible	55 (33.3)
	Incisors	10 (6.1)
	Canines	2 (1.2)
	Premolars	26 (15.8)
	Molars	17 (10.3)
Implant length (mm)	9	17 (10.3)
	11	57 (34.5)
	13	91 (55.2)
Bone quality	1. Very dense	14 (8.5)
	2. Dense	16 (9.7)
	3. Soft	88 (53.3)
	4. Very soft	38 (23.0)
	Not reported	9 (5.5)
Bone quantity	A. Ridge present	39 (23.6)
	B. Moderate resorption	60 (36.4)
	C. Advanced resorption	29 (17.6)
	D. Some resorption	12 (7.3)
	E. Extreme resorption	0
	Not applicable */not reported	25 (15.1)
Site type	Post-extractive	54 (32.7)
	Healing (25 h–12 weeks post-extraction)	6 (3.6)
	Healed (minimum 6 months)	105 (63.6)
Access to implant site	Full flap	61 (37.0)
	Minimally invasive flap (without releasing incisions)	51 (30.9)
	Flapless	39 (23.6)
	Not reported	14 (8.5)
Indication	Single crown	78 (47.3)
	Bridge	43 (26.1)
	Full-arch	44 (26.7)
Loading protocol	Immediate (<48 h)	50 (30.3)
	Early (48 h–3 months)	41 (24.8)
	Conventional (3–6 months)	14 (8.5)
	Delayed (>6 months)	14 (8.5)
	Submerged healing	44 (26.7)
	Not reported	2 (1.2)

* Not applicable was recorded for some extraction sockets.

**Table 2 jcm-11-04859-t002:** Selected surgical characteristics according to bone quality.

Bone Quality	Implant Site Preparation		Osseoshaping Instrument Insertion Torque			Final Implant Insertion Torque		
	*n* Assessed	Osteotomy Protocol (P1/P2/P3) *	*n* Assessed	Mean ± SD (Ncm)	Range (Ncm)	*n* Assessed	Mean ± SD (Ncm)	Range (Ncm)
Homogeneous compact bone (type 1)	14	6 (42.9%) 4 (28.6%) 4 (28.6%)	3	29.7 ± 7.7	21–35	14	49.9 ± 10.4	37–70
Thick layer of compact bone (type 2)	16	16 (100%) 0 0	9	34.3 ± 8.6	25–50	16	57.5 ± 13.6	27–71
Thin layer of cortical bone (type 3)	88	87 (98.9%) 1 (1.1%) 0	34	23.7 ± 10.3	6–45	88	44.7 ± 13.6	11–70
Low-density trabecular bone (type 4)	38	38 (100%) 0 0	21	12.1 ± 8.2	5–35	38	30.7 ± 17.5	9–75
Not reported	9	9 (100%) 0 0	6	20.3 ± 11.7	4–36	9	45.8 ± 18.1	10–70

* P1 (pilot drill, osseoshaping instrument, implant). P2 (pilot drill, osseoshaping instrument, dense bone osseoshaping instrument, implant). P3 (pilot drill, osseoshaping instrument, dense bone osseoshaping instrument, dense bone drill, implant).

## Data Availability

Restrictions apply to the availability of these data. Data were obtained from Nobel Biocare Services AG and are available with the permission of Nobel Biocare Services AG.

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
