# Peer review of "A Retrospective Observational Study Assessing the Clinical Outcomes of a Novel Implant System with Low-Speed Site Preparation Protocol and Tri-Oval Implant Geometry"

_jcm, 2022, doi:10.3390/jcm11164859_

Round 1
Reviewer 1 Report
The manuscript presented by the authors is of particular interest to the scientific community, as it describes the clinical results of a new surgical technique to create an operating site for an implant.
However, it is necessary to reformulate the title and remove (and from the text), the incorrect phrases of general vocabulary such as "Real World", or replace them with the corresponding accepted scientific terms.
3) I would like to clarify the formulation of the null hypothesis - why did the authors take as a basis the hypothesis that the new technique and system of implant trials are not inferior to classical well-established systems. Why are they not equivalent or even superior to existing ones?
To understand the advantages of the low-speed technique for creating an implant site, it is necessary to consider and describe the laboratory studies (in-vitro studies) performed with the corresponding bone site; this will allow one to evaluate the difference in the quantity / quality / speed of osteoingeration of the new approach compared to the analogues.
The authors use the novel technique with different loads on the implant (immediate, delayed), with different types of prosthetics (single tooth, bridge, full arch), installing implants immediately after removal, in a fresh socket, in a healed socket; however, they do not describe the difference in the results obtained depending on the above variations. This is a significant drawback and requires supplementation or data processing.
Who carried out the description and comparison of the results of the X-ray images? The manuscript states that all methods of X-ray examination, intraoral images, OPTG, and CBCT were used. For what purpose were all types of examination prescribed to patients? Alternatively, was any of the three used? How, in this case, can a comparison be made because the diagnostic resolution of these methods is significantly different? It is necessary to bring the protocol for radiological evaluation.
How the calibration of the operators (surgeons) was carried out in each of the three centers in which the study was conducted. Would you describe?
Reviewer 2 Report
1.Is the manuscript relevant and interesting? The article is relevant and interesting. 2.How original is the topic? The topic is current. 3.What does it add to the subject area compared with other published material? The authors have collected and analyzed original data. 4. Is the paper well written? Yes, the article is well written. 5. Is the text clear and easy to read? Minor English editing is required. 6. Are the conclusions consistent with the evidence and arguments presented? Yes, the conclusions consistent with the evidence and arguments presented. 7. Do they address the main question posed? Yes, the Authors addressed the main question posed. Other comments: · English language: Minor English editing is required. · Introduction: This section needs few improvements. For example, Authors may include a brief sentence at the beginning of this section regarding innovations in implant dentistry based on the following reference: <>. · · Materials and methods: This section has been properly prepared. · Results: This section has been properly prepared. · Discussion: What is the main theme that emerges from the authors' analysis? Is the study design a limitation? Please improve. · Conclusion: This section has been properly prepared. After making the indicated changes, the article may be suitable for publication after Editorial evaluation. Thanks for the opportunity to review this manuscript.
Author Response
Pleae see the attachment.

Round 2
Reviewer 1 Report
Dear authors! Thank you for the comprehensive editing work with the manuscript. It was a pleasure to read the report with the answers and comments to the first review. The revised version is significantly better and will be recommend for the publication.